# Impact of Process Parameters on the Quality of Deep Holes Drilled in Inconel 718 Using EDD

**DOI:** 10.3390/ma12142298

**Published:** 2019-07-18

**Authors:** Magdalena Machno

**Affiliations:** Rail Vehicles Institute, Mechanical Faculty, Cracow University of Technology, 31-155 Cracow, Poland

**Keywords:** EDD, drilling, deep hole, high aspect ratio hole, difficult-to-cut material, Inconel 718 alloy

## Abstract

Advanced engineering materials (e.g., nickel or titanium alloy) are being increasingly applied to produce parts of gas turbines in the aerospace industry. To improve the durability of these parts, many holes, with a length-to-diameter aspect ratio greater than 20:1, are created in their structure. The quality of the holes significantly affects the cooling process of the elements. However, it is challenging to machine materials by conventional methods. When machining a hole with a high aspect ratio, the major problem is effective flushing of the machining area, which can improve the hole’s surface integrity and dimensional accuracy. Consequently, the electro-discharge drilling (EDD) process is good alternative for this application. This paper presents the results of an analysis of the EDD of Inconel 718 alloy. An experiment was conducted to evaluate the impact of process parameters (pulse time, current amplitude, and discharge voltage) on the process’s performance (linear tool wear, taper angle, drilling speed, the hole’s aspect ratio, and surface roughness (*Ra* and *Rz*)). The results show that EDD provides us with the possibility to drill holes with an aspect ratio greater than 10:1. The results also demonstrate that holes with an aspect ratio greater than 10:1 and a small taper angle value have a significantly decreased quality of internal surface, especially at the bottom of the hole. This indicates that an insufficient amount of debris is removed from the bottom of the hole.

## 1. Introduction

Nickel-based superalloys, such as Inconel 718, have applications in a range of engineering areas, including aerospace, automobile, and medical engineering, because of their excellent mechanical and chemical properties (superior strength and good corrosive resistance) in high-temperature environments. These superalloys are most widely used in the aerospace industry in turbine blades, guide vanes, etc. [1,2,3]. Inconel 718 can be used within the temperature range of −290.15 to 973.15 K. The components of modern gas turbine engines need to resist temperatures higher than 2000 K (the melting point of nickel-based superalloys). Thus, to survive for a sufficiently long time under service conditions, superalloy materials require an additional internal and external cooling system. One of the techniques used to decrease the temperature of a component is the introduction of a large number of cooling channels inside the element. The cooling factor (a gas or liquid) that flows through the cooling holes cools the component down [4,5]. Gas turbine blades possess between 20,000 and 40,000 cooling holes that feature a diameter in the range of 0.3–5 mm and an aspect ratio greater than 20:1 (the depth-to-diameter ratio) [4,5,6]. To effectively decrease an element’s temperature, the holes should be manufactured with a high surface quality and a high dimensional accuracy that provides for a sufficient flow of cooling factor [7,8]. Furthermore, the large number of holes requires a technology that can efficiently manufacture cooling holes.

The production of a large number of cooling holes in superalloys is a complex task due to their small diameter and high aspect ratio. In some components, the holes are drilled at an angle of inclination. Moreover, nickel-based alloys are difficult to machine with conventional processes, as they have a strong tendency to weld and form a built-up edge [1,9,10].

At present, in the aerospace industry, one of the most effective methods for the drilling of deep holes with small diameters (less than 1 mm) is electrical discharge drilling (EDD) [11]. EDD is a well-known process with the advantage of being able to machine materials regardless of their hardness (e.g., nickel and titanium alloys, hard metals, superhard alloys, and ceramics) [12]. EDD’s additional advantages include a simple tooling process, a capacity to drill multiple holes simultaneously, a lack of burrs, and the ability to drill on angled surfaces [13]. The EDD process utilizes a thermal effect rather than mechanical force to remove material. It provides the possibility to drill using a longer tool electrode with a micrometer-size diameter. In EDD, the allowance is removed by the action of electrical discharges that occur between two electrodes in a narrow gap (~μm). One of the electrodes is the workpiece and the other one is a tool. The energy from a series of electrical discharges generated between the electrodes, which are immersed in a dielectric medium, erodes the material from the workpiece. The transformation of electrical energy into thermal energy leads to the vaporization and melting of the workpiece and the material electrode [14,15].

One of EDD’s limitations during deep hole drilling is caused by the accumulation of debris at the bottom of the hole, which contributes to the wearing of the reinforced side electrode and secondary discharges (e.g., arcing and short circuits). An especially high concentration of eroded particles occurs in the corner of the drilled hole [16]. These phenomena lead to high process instability, defects on the internal surface of the holes, holes with poor shape accuracy (in terms of hole taper and hole dilation), excessive tool wear (at the side and the end), and a low machining speed [12,13,14,17,18,19]. In the deep hole drilling process, a high-pressure pump for dielectric flushing (up to 25 MPa) and tube electrodes with inner flushing are used [20]. Flushing should be done in a sufficiently controlled manner. Flushing with a high dielectric pressure can sweep the plasma channel and lead to a decrease in the material removal rate [21]. To improve flushing, several methods can be applied, including internal flushing (flushing through single-and multi-channel electrodes) [13,16], external flushing, suction, different electrode movements (rotary, planetary) [18], vibration-assisted methods [12], simultaneous flushing with a vacuum-assisted debris removal system [21,22], and the use of tool electrodes coated with a material of low electrical conductivity [14].

In order to understand and to improve the EDD process, the influence of process parameters on the process’s performance was analyzed. Previous studies focused on the influence of major EDD parameters, such as peak current, voltage, pulse-on-time, exchanged power, flushing pressure, duty factor, frequency, and electrode rotation speed, on aspects of the process’s performance, such as the material removal rate, the tool wear rate, the overcut, and the taper, as well as a geometrical and dimensional analysis of the holes [23,24,25,26].

In [24], the effect of electro-discharge micromachining parameters (peak current, pulse-on-time, flushing pressure, and duty factor) on the process’s performance (material removal rate, tool wear rate, overcut, and taper) while machining Ti-6Al-4V alloy was investigated. The results of experimental investigations showed that the pulse time was the process parameter with the most influence on the material removal rate, overcut, and taper, whereas the peak current had the maximum percentage of contribution to tool wear. In [23], the impact of machining process parameters (peak current, pulse-on-time, frequency, and spindle rotation speed) on the geometrical characteristics (overcut and taper rate) of micro-holes and the machining performance was examined. Peak current was found to have a strong influence on the geometrical characteristics of the micro-holes. An increase in peak current was found to contribute to an increase in the overcut and taper rate. The results of the analysis showed that spindle rotation speed had an insignificant effect on the machining performance. The authors in [25] analyzed the effect of process parameters (peak current, voltage, and exchanged power) on aspects of the process’s performance, including the geometrical characteristics of the hole (difference between the hole top’s diameter and the electrode’s nominal diameter (*DOC*) and taper rate (*TR*)), the material removal rate (*MRR*), and the tool wear rate (*TWR*). The results showed that the tool with a diameter of 300 µm had an influence on the highest number of process indicators, including *DOC*, *MRR*, and *TWR*. The *MRR* and the *TWR* were found to be more affected by variation in the parameters than the geometrical indicators (*DOC* and *TR*).

It is worth noting that, in the EDD process, both the process parameters and the electrode characteristics (i.e., the workpiece material’s and the electrode tool’s characteristics) have an influence on aspects of the process’s performance (such as the material removal rate, the tool wear rate, the diametric overcut, and the taper rate). In [27], the authors adopted an index to take into account both the process parameters and the properties of the work material and the electrode material during electrical discharge micro-drilling. The “Material and Technological Index” (*MTI*) for each kind of tool material (such as brass tungsten carbide) was defined while taking into account EDD process parameters and electrical and thermal properties of the workpiece and electrode material. The defined indexes were used to demonstrate the influence of the electrode properties and the insignificant influence of the workpiece characteristics.

In addition, researchers have developed mathematical models based on an analysis of the effects of input parameters (current amplitude, time of the impulse, duty cycle, voltage) on the process’s performance using various optimization techniques (e.g., roughness surface methods (*RSM*), ANOVA techniques, fuzzy logic, artificial neutral network (*ANN*), Taguchi analysis) [13,28,29]. The selection of process parameters with appropriate values plays an important role in obtaining a high material removal rate, low tool wear, high surface quality (a low surface roughness parameter), and holes with high shape accuracy [30,31,32].

In [30], the authors developed an optimization method to achieve a higher material removal (MRR) rate with the desired hole accuracy and surface finish in EDD. In the first step, the influence of the factors’ peak current, duty factor, and electrode rotation speed, the higher-order effects of the pulse-on-time, and the interaction effects between peak current and duty factor and peak current and electrode rotation, on the MRR were analyzed. In the next step, the desirability function approach (DFA) was chosen to optimize the process parameters. The optimized parameters for the maximum material removal rate produced surface roughness values of 3 µm and 3.5 µm. In [24], a higher material removal rate and a lower tool wear rate, overcut, and taper were obtained on the basis of a Taguchi analysis of the optimal combination of EDD process parameters (peak current, pulse-on-time, flushing pressure, and duty factor), which were determined to be: (a) 1.5 A/10 μs, 0.5 kg cm^−2^/95%; (b) 0.5 A/1 μs/0.3 kg cm^−2^/60%; (c) 0.5 A/1 μs/0.1 kg cm^−2^/60%; and (d) 1.5 A/10 μs/0.5 kg cm^−2^/95%, respectively.

The phenomenon occurring in the machining area between electrodes are still weakly recognized as what prevents an appropriate selection of process parameter values. The need for improvement of the EDD process, especially drilling high aspect ratio holes, is present. Further experimental research should be carried out to improve the efficiency of the process, including high material removal rate, low tool wear, satisfied dimensional accuracy, and quality of drilled holes.

Herein, this study presents the results of the experiments on electrical discharge drilling of Inconel 718 alloy using a tube electrode. During the experiment, the impact of the process parameters pulse time, current amplitude, and discharge voltage on the process’s performance was studied. The process’s performance was analyzed in terms of linear tool wear, taper angle, drilling speed, the aspect ratio of holes, and the surface roughness of holes (*Ra* and *Rz*). To study the relationship between process parameters and performance criteria, ANOVA techniques were applied. The aim of this study is to improve the EDD process to enable us to drill holes with a high aspect ratio and a satisfactory accuracy and to investigate the influence of process parameters on the process’s performance. In the drilling experiment, a sample consisting of two parts was used. The drilling was carried out at the junction of the two parts of the sample. Once the two parts of the sample were separated, an analysis of the dimensional accuracy and the geometric characteristics of the internal surface of the holes was carried out. This approach helps us to understand the phenomenon that occurs at the bottom of a hole during the EDD process.

## 2. Materials and Methods

### 2.1. Material

Inconel 718 alloy was used as the workpiece material. The chemical composition and the main mechanical properties of this material are presented in Table 1 and Table 2, respectively. A special sample, consisting of two parts, was designed and produced for the purpose of the experiment. The drilling of the holes was carried out at the junction of the two parts (Figure 1a). After drilling, the sample parts were separated to provide us with the possibility to analyze the dimensional accuracy of holes and the quality of the inner surface. To avoid problems with through drilling, an additional technological pad was applied on the underside of the sample. In order to minimize the impact of electrode vibrations and clamping eccentricity on the drilling process, an electrode guide system was employed, as shown in Figure 1b.

### 2.2. Experiment Design

The electrical discharge drilling was carried out on the experimental test stand shown in Figure 2. The purpose of the tests was to examine the impact of selected machining parameters on the dimensional accuracy of the hole, the surface quality, the machining efficiency, and tool electrode wear. Table 3 presents the data on the drilling process and Table 4 presents the adopted ranges of values. The experiments were performed according to the theory of the experiment using a three-level rotatable research plan that included 20 experimental tests with six repetitions in the center of the research plan. The results of the experiments are shown in Table 5. The ANOVA techniques were applied to investigate the relationship between process parameters and input parameters.

The following constant parameters were assumed: Initial interelectrode gap (*S*_0_ = 50 μm), inlet dielectric fluid pressure (*p_in_* = 8 MPa), rotational speed of the clamp and the electrode (*n* = 400 rpm), drilling time of each test (*t_d_* = 45 min), pulse-off-time (*t_off_* = *t_i_*), dimensions and material of the tube electrode (single-channel; outer diameter: 1 mm, inner diameter: 0.3 mm; made of copper) (Figure 3a), and deionized water with a low electrical conductivity as a dielectric fluid. Before each experiment was started, the temperature (*T* = 297.15–318.15 K) and electrical resistivity (*κ* = 2–3.9 µS/cm) of the deionized water were measured. The dielectric fluid was flushed down the interior hole of the tube in order to remove eroded particles (Figure 3b).

The linear tool wear (*TW)* was calculated according to the following formula:
*TW* = (*h_t_*/*h_h_*)∙100%,(1)
where *h_t_* is the shortening of the electrode and *h_h_* is the hole depth. The taper angle (*tap_α_*) (Figure 4) is calculated from the following equation:
*tap_α_* = (*D_in_* − *D_out_*)/2*h_h_*,(2)
where *D_in_* is the average top diameter and *D_out_* is the average bottom diameter.

The drilling speed (*v*) was calculated from the following equation:
*v* = *h_h_*/*t_d_*,(3)
where *t_d_* is the drilling time. The aspect ratio (*AR*) was given by the following equation:
*AR* = *h_h_*/*D_average_*,(4)
where *D_average_* = (*D_in_* + *D_out_*)/2 is the average diameter of the hole.

The average diameters along the hole’s depth were measured on two parts of the sample (five measurements were made for each diameter). The measurements were performed using a K-401 stereo microsope with a Common Main Objective (CMO) Infinity optical system and a Moticam 2300 digital camera (Richmond, Canada). The measurements of the diameters were carried out using a MoticImages Plus system. The difference between the appropriate diameters of the two sample parts was in the range of 20–30 µm, which led us to assume that the drilled holes were symmetrical.

The average values of surface roughness (*Ra* and *Rz*) were measured using a Talysurf Intra 50 profilometer (Taylor Hobson, Leicester, UK). In order to perform the surface roughness measurements, a measuring tip with a rounding radius of 2 µm was used. The measurements were made along the direction of the hole’s depth (parallel to the measuring axis of the hole). A measurement speed of 1 mm/s was used. For the measurements in the two-dimensional (2D) system, the resolution of the X axis was equal to 1 µm and five elementary sections 0.8 mm in length were applied.

## 3. Results and Discussion

### 3.1. The Impact of Input Parameters on the Dimensional and Shape Accuracy of Holes

To investigate the influence of machining parameters on the process’s performance, an ANOVA analysis was employed. The obtained regression equations *TW*(*t_i_*, *I*, *U*), *tap_α_*(*t_i_*, *I*, *U*), *v*(*t_i_*, *I*, *U*), *AR*(*t_i_*, *I*, *U*), *Ra*(*t_i_*, *I*, *U*), and *Rz*(*t_i_*, *I*, *U*) are described as Equations (5)–(10), respectively.
*TW*(*t_i_*, *I*, *U*) = 36.195 − 0.0308 × *t_i_* − 3.629 × *I* − 0.0376 × *U* + 0.0073 × *t_i_* × *I*,(5)
*tap_α_*(*t_i_*, *I*, *U*) = 0.0778 + 4.0707 × 10^−5 ^× *t_i_* − 0.0124 × *I* − 2.461 × 10^−4^ × *U* − 3.218 × 10^−8 ^× *t_i_*^2^,(6)
*v*(*t_i_*, *I*, *U*) = 19.424 − 0.0012 × *t_i_* − 14.344 × *I *+ 0.0688 × *U* + 2.307 × *I*^2^,(7)
*AR*(*t_i_*, *I*, *U*) = 24.889 − 0.0041 × *t_i_* − 21.2 × *I *+ 0.162 × *U* + 3.637 × *I*^2^,(8)
*Ra*(*t_i_*, *I*, *U*) = −2.237 + 0.0052 × *t_i_* + 0.765 × *I *+ 0.0075 × *U* − 3.706 × 10^−6^ × *t_i_*^2^,(9)
*Rz*(*t_i_*, *I*, *U*) = −71.197 + 0.0251 × *t_i_* + 19.183 × *I *+ 0.727 × *U* − 1.328 × 10^−5 ^× *t_i_*^2^ − 0.169 × *I *× *U*.(10)

An analysis of the results shows that the average top diameters *D_in_* were greater than the average bottom diameters *D_out_*. This may be related to simultaneous side wall tool wear during the drilling process. The larger side gap in the hole’s bottom area could not provide the conditions under which electrical discharges occur. However, the average value of the bottom diameter increased as the current amplitude *I* increased (in spite of the hole’s conical shape). Figure 5 and Figure 6 present the dimensional shape of the holes for extreme values of applied current amplitude. An increase in current caused a higher amount of energy to be transferred in a single discharge; thus, a higher amount of material was removed. For *I* > 4 A, the drilled holes were characterized by a higher aspect ratio (*AR* > 10). A higher *AR* prevented the removal of debris (due to insufficient flushing in the machining area’s inter-electrode gap and the side gap at the bottom of the hole) and strengthened the accumulation of debris. The conditions in the machining area could produce secondary discharges that would expand the bottom diameter. In this case, an increase in the bottom diameter could produce a decrease in *tap_α_*. At the maximal value of applied current *I* = 4.65 A (*U* = 100 V and *t_i_* = 550 µs), the taper angle of the hole was reduced to the minimum value (about zero) (Figure 7a).

The results showed that the drilling speed (*v*) depended mainly on the discharge voltage *U* and the current amplitude *I*. The pulse time (*t_i_*) had an insignificant effect on the drilling speed. The amount of removed material depended on the energy in a single discharge. A higher *U* and *I* would cause a higher discharge energy and an increase in the drilling speed. In addition, a higher discharge voltage (*U* = 100–120 V) could cause bailing and the evaporation of water. Gas bubbles could hinder the dielectric flow and the removal of debris. The highest values of drilling speed (*v* = 10 µm/s) and aspect ratio (*AR* = 23) were obtained for the machining parameters *U* = 120 V, *t_i_* = 282 µs, and *I* = 4.32 A. The analysis of the results showed a decrease in the taper angle (*tap_α_* = 0.006) in the case where a higher *AR* > 10 was obtained. This confirmed that the accumulation of debris was strengthened and a higher number of secondary discharges occurred. Since the same drilling time (*t_d_* = 45 min) was applied in each experiment, it was concluded that the drilling performance influenced the obtained *AR* values (Figure 7b).

Table 6 and Table 7 present the ANOVA results (where DF is degrees of freedom, Seq SS is sums of squares, Adj SS is the adjusted sums of squares, and Adj MS is the adjusted means squares).

An increase in the pulse time and current amplitude led to a decrease in the taper angle (*tap_α_*) (Figure 8). In the case where deionized water was applied as a working fluid, the electrical discharges were accompanied by electrochemical dissolution. Deionized water can be treated as a dielectric fluid and a weak electrolyte that contributes to material removal by the simultaneous interaction of electrochemical dissolution and electrical discharges in single discharge [33]. The analysis of the voltage and current waveforms of a single discharge showed that the time duration of the electrical discharge constituted approximately 50–60% the entire single pulse time (regardless of the time duration applied to the pulse). For this reason, a higher value of pulse time *t_i_* = 999 µs (*U* = 100 V, *I* = 3.83 A) extended the time during which electrical discharges and abnormal discharges occurred. However, a higher value of applied current amplitude caused a larger amount of material to be removed, which resulted in a higher accumulation of debris. Figure 9 and Figure 10 show the obtained hole shapes for extreme values of applied *t_i_*. In the case where the pulse time was extended, the average bottom diameter was higher and the taper angle was decreased. The dimensional accuracy of the hole was improved in comparison to the case where a shorter pulse time was applied.

Table 8 presents the ANOVA results.

The amount of linear tool wear was found to be in the range of *TW* = 13–20%. A decrease in linear tool wear took place for higher values of discharge voltage and a longer pulse time (Figure 11). Based on the results analysis in [5,34], the application of deionized water produced a decrease in linear tool wear with a simultaneous increase in the process’s performance. In [35], the results analysis revealed that a sufficiently high value of applied discharge voltage and a long pulse time enabled material to be removed by electrochemical dissolution and electrical discharges in a single impulse. The voltage between the electrodes should be increased gradually. This could provide us with more stable machining conditions and an improvement in the flushing gap. The measured diameter of the tool tip after drilling was approximately 20% and 10% lower than the nominal value for an applied *t_i_* = 550 µs and an applied *t_i_* = 999 µs (*U* = 100 V and *I* = 3.83 A), respectively. Additionally, a higher applied discharge voltage (*U* = 100–120 V) reinforced electrochemical reactions, resulting in an increase in *AR* (Figure 7b). The ANOVA showed that current amplitude had an insignificant effect on linear tool wear. Table 9 presents the ANOVA results.

### 3.2. The Geometric Structure of Holes

The average values of the surface roughness parameters *Ra* and *Rz* increased as the current amplitude increased and the time of the impulse was extended (Figure 12a,b). In [35], the results analysis showed that an increase in the values of the parameters’ current and pulse duration led to an increase in the number of bubbles in the machining area, which decreased the area between the electrodes. This condition is favorable for the bubbles to break down the voltage and to initiate the occurrence of sparks and abnormal discharges (between debris and the side gap). A large number of bubbles can cause difficulties when flushing the machining area and can decrease the stability of the machining conditions. The machining parameter values *U* = 100 V, *t_i_* = 550 µm, and *I* = 3.83 A produced average values of *Ra* and *Rz* equal to 3.17 µm and 19.82 µm, respectively. Increasing the current amplitude to 4.65 A (*U* = 100 V, *t_i_* = 550 µs) caused the average values of *Ra* and *Rz* to increase by approximately 23–30%. Extending the pulse time to *t_i_* = 999 µs (*U* = 100 V, *I* = 3.83 A) caused the average values of *Ra* and *Rz* to increase approximately two-fold in comparison to an applied pulse time of *t_i_* = 100 µs. For the minimal value of applied pulse time *t_i_* = 100 µs (*U* = 100 V, *I* = 3.83 A), an improvement in the surface quality occurred. The obtained values of the surface roughness parameters were *Ra* = 1.59 µm and *Rz* = 11.59 µm. Table 10 and Table 11 present the ANOVA results.

### 3.3. Quality of the Internal Surface of Holes

The internal surface of the drilled holes on both parts of the sample darkened along the hole’s depth (Figure 13). The darker colored surface at the bottom of the hole was strengthened, especially with an aspect ratio of *AR* ≥ 10. The accumulation of debris and an excessive occurrence of abnormal discharges (significantly higher heating of the gap medium) produced a change in the machined surface properties.

The darker machined surface at the bottom of the hole was subjected to a qualitative analysis using a scanning electron microscope (SEM) with an energy dispersive spectroscopy system (EDS) produced by JEOL Ltd. (Tokyo, Japan), as shown in Figure 14. An analysis of the results revealed a volume percent increase in elements such as Ni, Fe, Cr, and O (Figure 15a). These elements constitute the alloy elements of Inconel 718.

The analysis performed for the three points in Zone I indicated differences in the volume percent of the main alloy elements. In Zone 2 (Figure 14), the percentage involvement of Ni was about 67% (in the chemical composition of Inconel 718, the maximum percentage of Ni is 55%) (Figure 15c). In Zone 3, the percentage involvement of Cr was about 43% (in the chemical composition of Inconel 718, the maximum percentage of Cr is 21%) (Figure 15d). Unremoved and re-solidified debris could be found on the hole’s internal surface. In addition, in Zones 1 and 3 (Figure 15b,d), oxygen was identified (an average value of about 17 wt.%). This resulted when using deionized water as a dielectric fluid.

Figure 16 shows SEM images of three surfaces from the bottom, center, and top areas of the holes. A significant number of microcracks was observed at the bottom of the hole with an aspect ratio of *AR* ≥ 10. The electrode rotation speed contributed to the electrode’s striking with high force at the eroded particles on the hole’s bottom surface. From the observed microcracks, we deduced that spalling took place under the material-removal mechanism.

The experimental research, including the analysis of the working fluid’s flow through the tubular electrode, showed that the flow rate of dielectric fluid, based on the volume flow rate measurements, amounted to approximately 30 m/s (the volume flow rate *Q_m_* = 2.57 × 10^−6^ m^3^/s and the cross-sectional area of the electrode’s internal diameter *S* = 9.14 × 10^−8^ m^2^). The estimated Reynolds number (*Re* > 5000) indicated a turbulent flow. A turbulent flow can contribute to the creation of whirls at the bottom of the hole. The presence of whirls can prevent debris from being removed from the bottom of the hole.

## 4. Conclusions

This study concerns the drilling of high aspect ratio holes in Inconel 718 alloy using EDD. The dimensional and shape accuracy of holes plays an important role in providing an appropriate flow of cooling factor, e.g., in turbine blades. For this reason, the technology used to manufacture holes should enable a high dimensional accuracy and good geometric structure (low values for the surface roughness parameters *Ra* and *Rz*). The number of the manufactured holes in parts of an aeroplane’s engine can be greater than 20,000. For this reason, the technology used to manufacture the holes should be efficient. EDD provides us with the possibility to manufacture holes in difficult-to-cut materials, such as nickel-based alloy. However, there are disadvantages to using the EDD process that constitute limitations when drilling microholes.

The results analysis indicated that electrical discharge drilling is a good alternative process for the manufacturing of high aspect ratio holes in Inconel 718 alloy. It obtained a maximal aspect ratio of 23 (for the process parameters *U* = 120 V, *t_i_* = 282 µs, and *I* = 4.32 A) and a drilling speed of approximately 10 µm/s. The results analysis enabled us to draw the following conclusions:The drilled holes had a conical shape; however, a high value of applied current amplitude (*I* = 4.65 A, *U* = 100 V, *t_i_* = 550 µs) decreased the hole’s taper angle (*tap_α_*) to approximately zero. This can result from a higher amount of accumulated debris at the hole’s bottom and the occurrence of secondary discharges that expand the hole’s output diameter.An increase in pulse time led to a decrease in the taper angle. This can result from an extended time of electrical discharge and secondary discharge. A lower taper angle value produced an increase in the accuracy of the hole.The application of deionized water as a working fluid can contribute to the removal of material by simultaneous electrochemical reactions and electrical discharges. A longer impulse time (*t_i_* = 1000 µs) extended the electrochemical reactions. Additionally, a higher discharge voltage (*U* = 100–120 V) reinforced the electrochemical reactions. This produced a decrease in tool wear and an increase in the hole’s aspect ratio.The minimal values of the surface roughness parameters, *Ra* = 1.59 µm and *Rz* = 11.59 µm, were obtained for the process parameters *t_i_* = 100 µs (the minimal applied value), *U* = 100 V, and *I* = 3.83 A.A darker color along the hole’s depth and a significant number of microcracks at the bottom of the hole were observed (especially for holes with an aspect ratio greater than 10). This could have resulted from the accumulation of debris and the occurrence of secondary discharges in the machining area (the inter-electrode gap and the side gap at the bottom of the hole) and from the removal of material by spalling.In those cases where the obtained hole had a conical shape (*D_in_* > *D_out_*) but a high aspect ratio (*AR* > 10), accumulation of debris at the bottom of the hole took place and this had an impact on the geometric structure of the hole.Insufficient flushing efficiency constitutes a limitation of EDD for deep hole drilling. To address this limitation, a special technology that supplies or suctions out working fluid could be applied.Further experiments should include additional process parameters for the properties of the working fluid, e.g., temperature or density.

## Figures and Tables

**Figure 1 materials-12-02298-f001:**
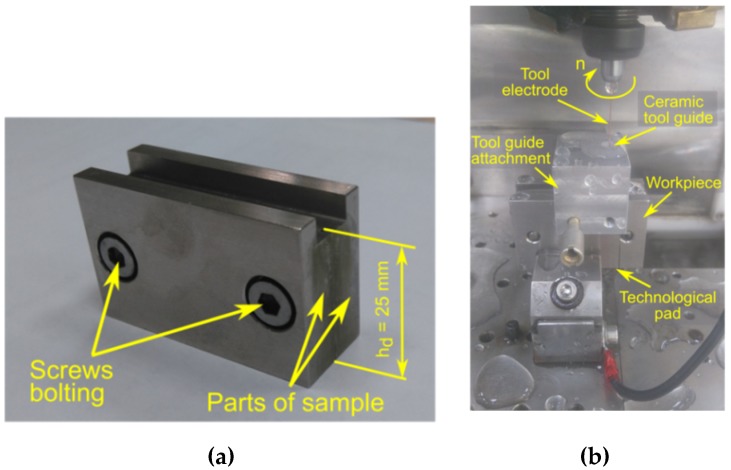
A photograph of the sample. (**a**) *h_d_*—Maximal drilling depth; (**b**) the experimental setup of the electrode guiding system.

**Figure 2 materials-12-02298-f002:**
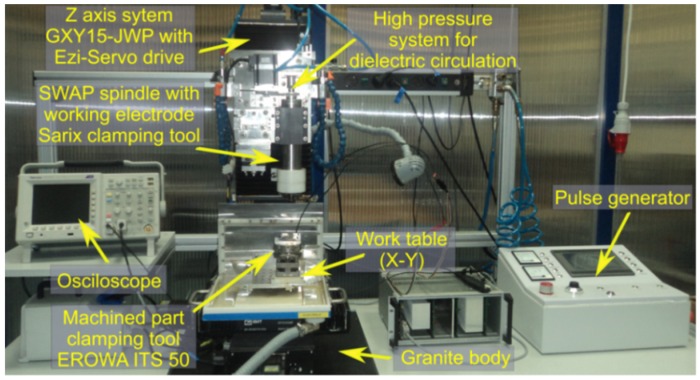
A photograph of the experimental setup.

**Figure 3 materials-12-02298-f003:**
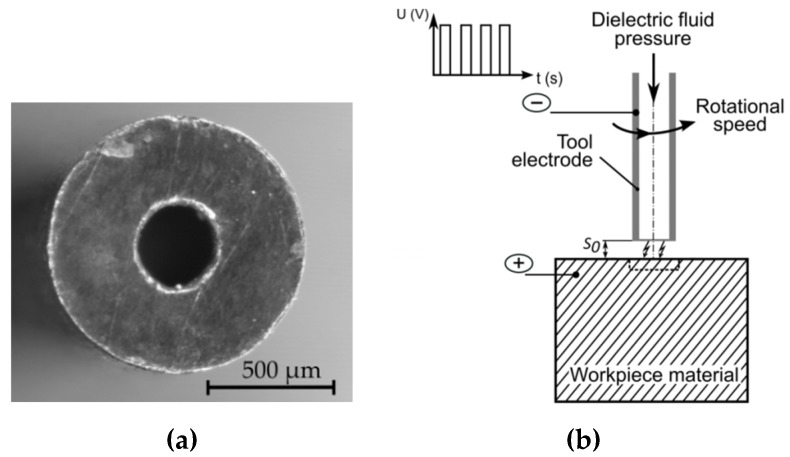
(**a**) The tip of the tool electrode; (**b**) the scheme of the electrical discharge drilling (EDD) process.

**Figure 4 materials-12-02298-f004:**
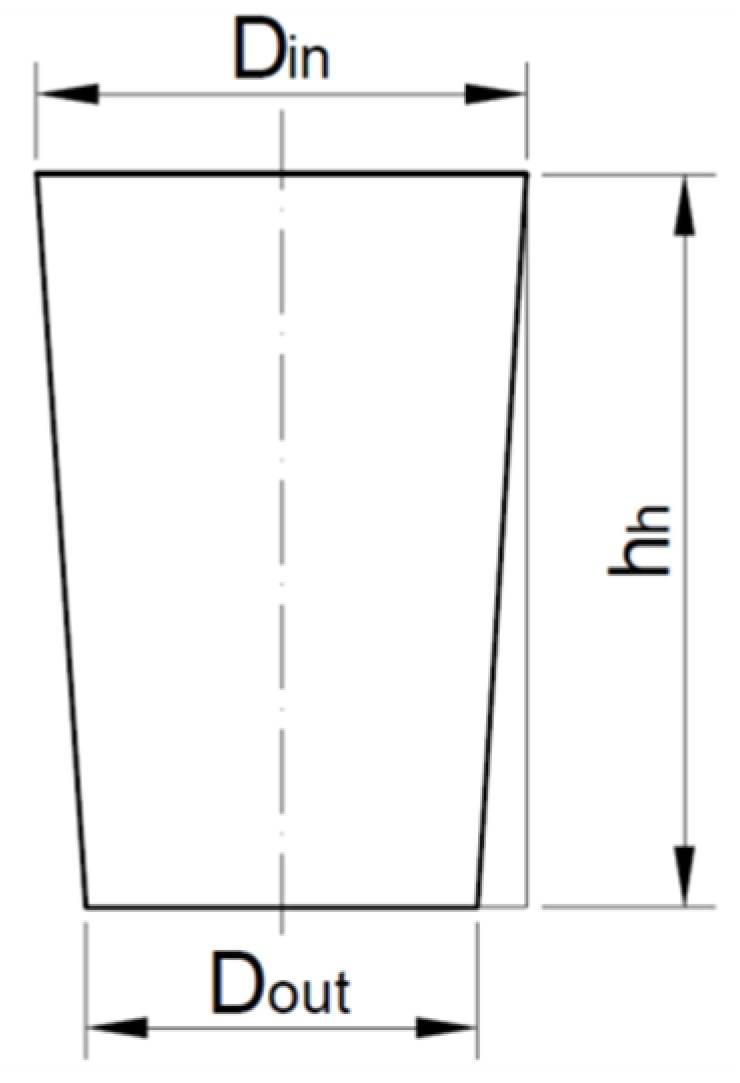
Measurement of the diameter accuracy and the taper angle (*tap_α_*).

**Figure 5 materials-12-02298-f005:**
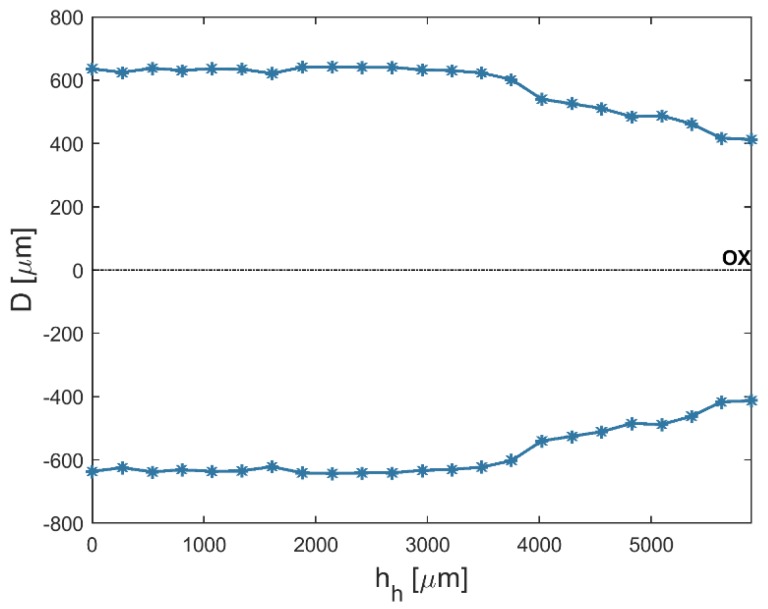
The profile of the drilled hole for the minimum value of applied current amplitude *I* = 3 A, *U* = 100 V, *t_i_* = 550 μs, and aspect ratio *AR* = 6.

**Figure 6 materials-12-02298-f006:**
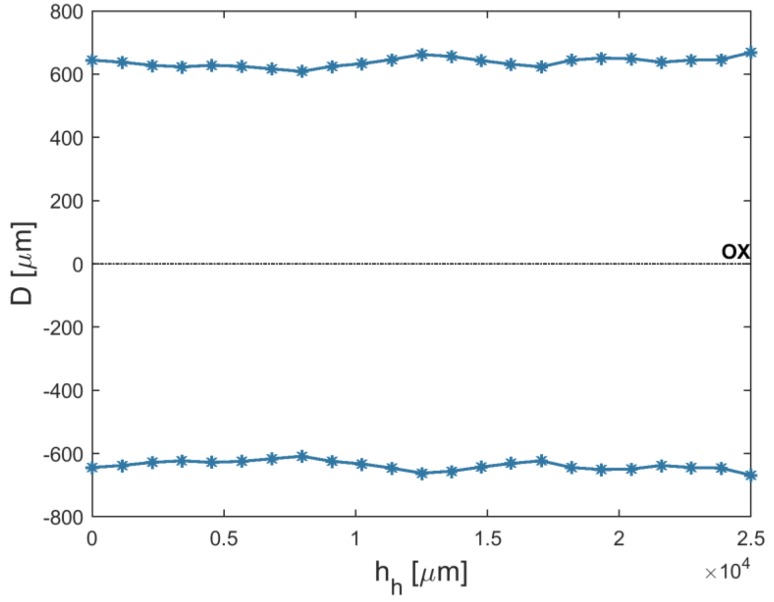
The profile of the drilled hole for the maximum value of applied current amplitude *I* = 4.65 A, *U* = 100 V, *t_i_* = 550 μs, and *AR* = 19.

**Figure 7 materials-12-02298-f007:**
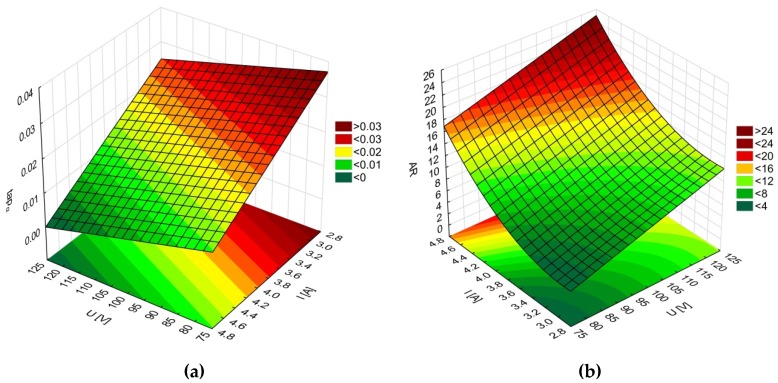
Relationship between the current amplitude (*I*) and the discharge voltage (*U*) and the process’s performance: (**a**) taper angle (*tap_α_*); (**b**) aspect ratio (*AR*); *t_i_* = 550 µs.

**Figure 8 materials-12-02298-f008:**
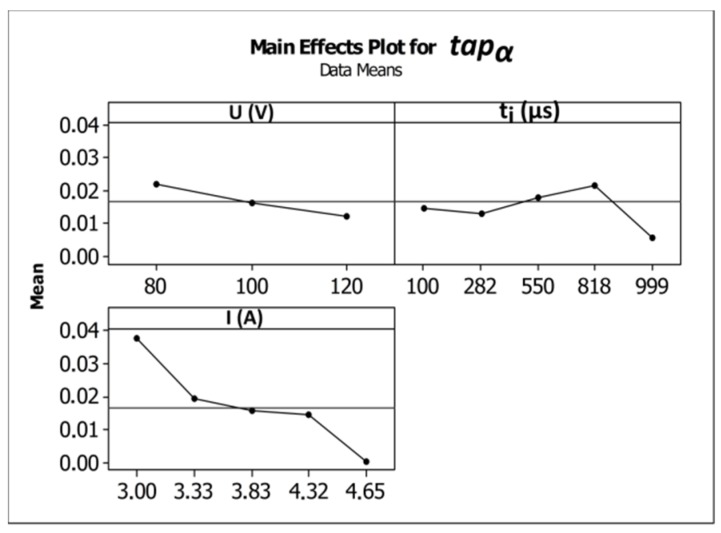
Influence of the machining parameters (*U*, *t_i_*, *I*) on the values of the taper angle (*tap_α_*).

**Figure 9 materials-12-02298-f009:**
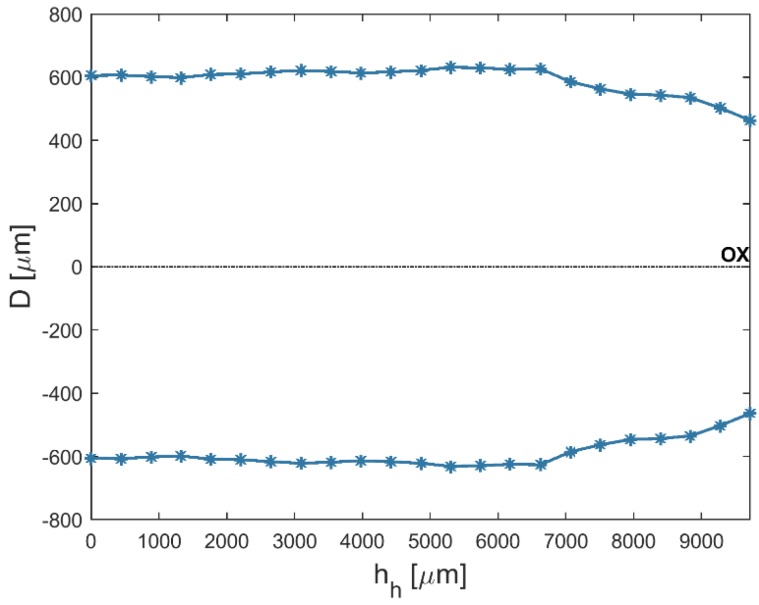
The profile of the drilled hole for the minimum value of applied pulse time *t_i_* = 100 μs, *U* = 100 V, *I* = 3.83 A, and *tap_α_* = 0.015.

**Figure 10 materials-12-02298-f010:**
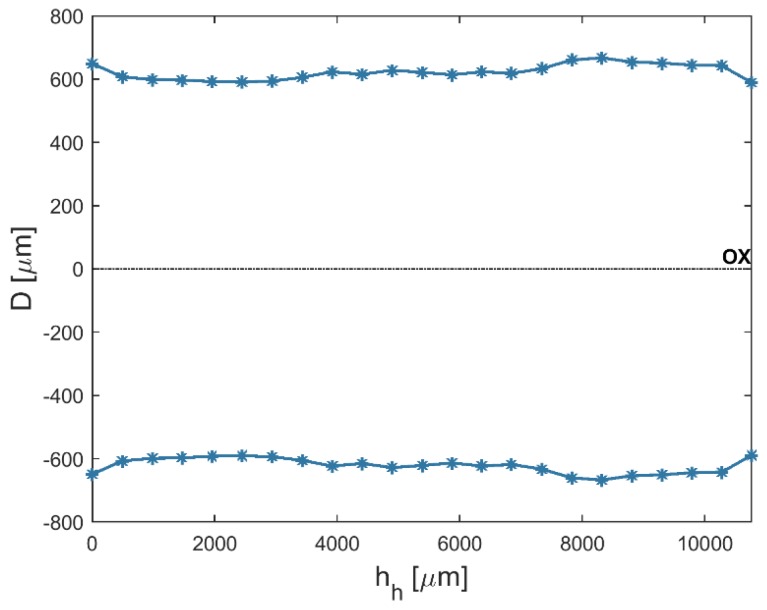
The profile of the drilled hole for the maximum value of applied pulse time *t_i_* = 999 μs, *U* = 100 V, *I* = 3.83 A, *tap_α_* = 0.006.

**Figure 11 materials-12-02298-f011:**
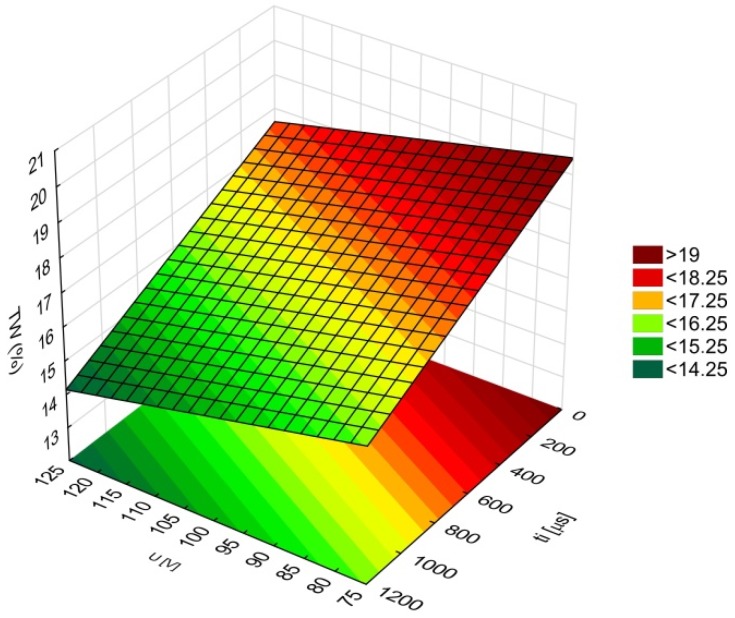
Relationship between linear tool wear (*TW*), and discharge voltage (*U*), and pulse time (*t_i_*); *I* = 3.83 A.

**Figure 12 materials-12-02298-f012:**
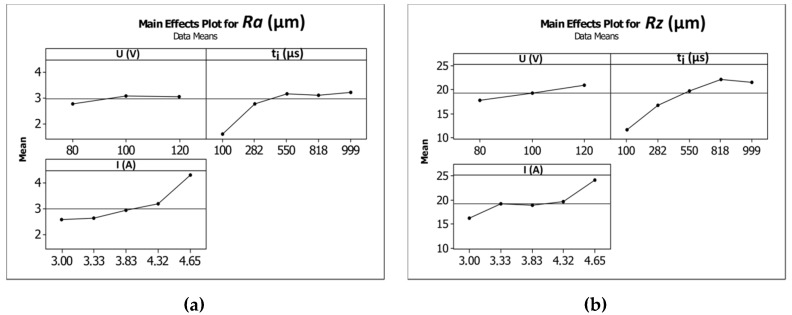
Influence of the machining parameters (*U*, *t_i_*, *I*) on the average values of surface roughness parameters. (**a**) *Ra* and (**b**) *Rz*.

**Figure 13 materials-12-02298-f013:**
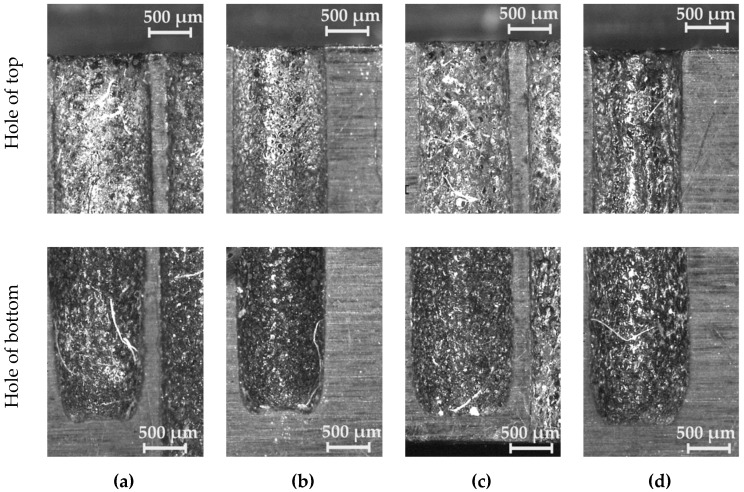
Images of the internal surface of the hole for the machining parameters. (**a**) *U* = 80 V, *t_i_* = 818 µs, *I* = 3.33 A, and *AR* = 6; (**b**) *U* = 100 V, *t_i_* = 100 µs, *I* = 3.83 A, and *AR* = 10; (**c**) *U* = 100 V, *t_i_* = 282 µs, *I* = 4.32 A, and *AR* = 23; (**d**) *U* = 100 V, *t_i_* = 550 µs, *I* = 3.83 A, and *AR* = 12.

**Figure 14 materials-12-02298-f014:**
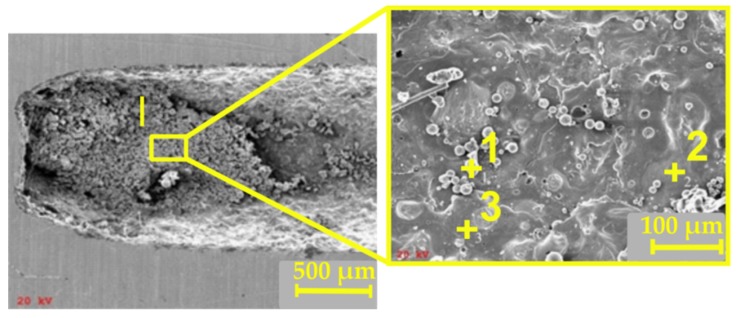
Scanning electron microscopy (SEM) images of the internal surface of the hole subjected to a qualitative analysis in Zone I. Machining parameters: *U* = 80 V, *t_i_* = 818 µs, *I* = 4.32 A.

**Figure 15 materials-12-02298-f015:**
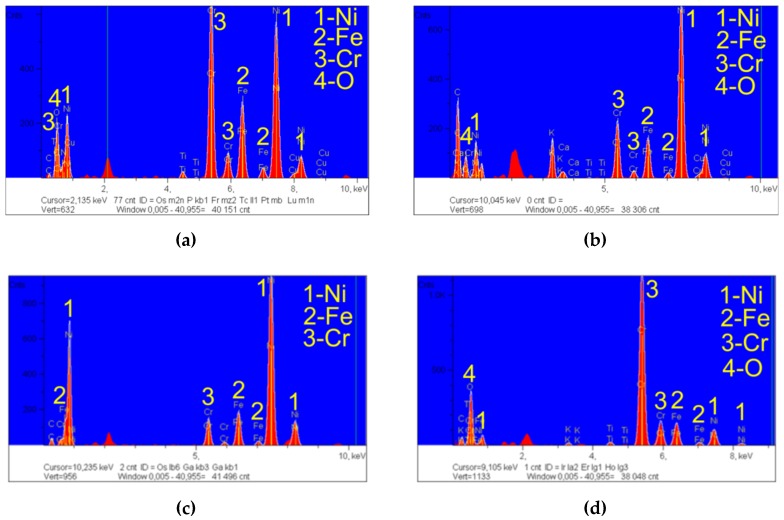
The chemical composition of the internal surface of the hole as determined by EDS. (**a**) In Zone I; (**b**) at point 1; (**c**) at point 2; and (**d**) at point 3.

**Figure 16 materials-12-02298-f016:**
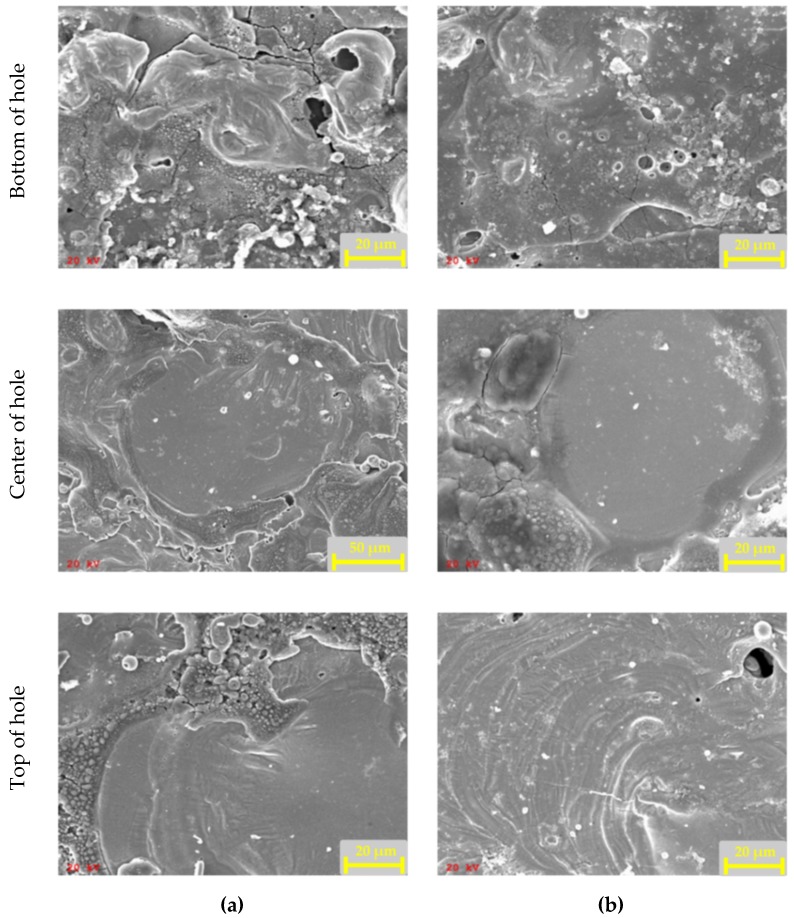
SEM images of selected internal surfaces of the hole: Bottom, center, and top. Machining parameters: (**a**) *U* = 80 V, *t_i_* = 818 µs, *I* = 4.32 A, and *AR* = 10; (**b**) *U* = 80 V, *t_i_* = 818 µs, *I* = 3.33 A, and *AR* = 6.

**Table 1 materials-12-02298-t001:** The chemical composition of Inconel 718 (wt.%).

Ni	Cr	Fe	Nb	Mo	Ti	Al	Co	Mn	C	Si	P
50.0–55.0	17.0–21.0	Balance	4.75–5.5	2.8–3.3	0.65–1.15	0.2–0.8	<1.0	<0.35	<0.08	<0.35	<0.015

**Table 2 materials-12-02298-t002:** The thermal properties of Inconel 718.

Density (kg/m^3^)	8190
Heat Capacity (J/(kgK))	0.2T (°C) + 421.7
Thermal Conductivity (W/(mK))	0.015T (°C) + 11.002
Melting Range (K)	1563.15–1623.15

**Table 3 materials-12-02298-t003:** Machining parameters.

Input Parameters	Output Parameters
Pulse time, *t_i_* (μs)	Linear tool wear (*TW*) (%)
Current amplitude, *I* (A)	Taper angle (*tap_α_*)
Discharge voltage amplitude, *U* (V)	Drilling speed (*v*) (μm/s)
	Aspect ratio hole (*AR*)
	Surface roughness parameters (*Ra*, *Rz*) (μm)

**Table 4 materials-12-02298-t004:** The process parameters and their levels.

Coded Parameter	Real Parameter	Level				
		1	2	3	4	5
A	*t_i_* (µs)	100	282	550	818	999
B	*I* (A)	3	3.33	3.83	4.32	4.65
		1	2			3
C	*U* (V)	80	100			120

**Table 5 materials-12-02298-t005:** The research plan and the results of the experiments.

Experiment Number	A	B	C	*t_i_* (µs)	*I* (A)	*U* (V)	*TW* (%)	*tap_α_*	*v* (μm/s)	*AR*	*Ra* (μm)	*Rz* (μm)
1	2	2	1	282	3.33	80	18.70	0.019493	2.88	7	2.15	14.27
2	2	4	1	282	4.32	80	16.61	0.015277	4.85	12	3.07	16.26
3	4	2	1	818	3.33	80	16.00	0.032308	2.65	6	2.58	16.62
4	4	4	1	818	4.32	80	20.22	0.024724	4.00	9	3.19	22.25
5	2	2	3	282	3.33	120	15.73	0.010286	5.91	15	2.81	16.74
6	2	4	3	282	4.32	120	17.02	0.006119	9.79	23	2.96	19.60
7	4	2	3	818	3.33	120	14.02	0.016559	4.65	11	3.00	29.14
8	4	4	3	818	4.32	120	16.81	0.012084	7.18	17	3.56	20.54
9	3	3	1	550	3.83	80	16.60	0.017814	3.65	9	2.76	19.14
10	3	3	3	550	3.83	120	17.02	0.015348	4.25	11	2.92	18.36
11	1	3	2	100	3.83	100	19.30	0.014567	3.60	10	1.59	11.59
12	5	3	2	999	3.83	100	13.62	0.005594	3.99	9	3.19	21.54
13	3	1	2	550	3.00	100	18.51	0.037792	2.18	6	2.57	16.18
14	3	5	2	550	4.65	100	16.36	0.000002	9.66	19	4.30	24.17
15	3	3	2	550	3.83	100	17.19	0.011059	5.34	12	2.67	18.15
16	3	3	2	550	3.83	100	18.04	0.019375	4.25	10	4.13	21.74
17	3	3	2	550	3.83	100	19.44	0.014754	4.61	10	3.26	18.43
18	3	3	2	550	3.83	100	14.82	0.018340	4.57	11	2.97	18.67
19	3	3	2	550	3.83	100	16.80	0.025359	3.92	10	3.40	16.14
20	3	3	2	550	3.83	100	16.36	0.016618	6.74	12	2.59	15.77

**Table 6 materials-12-02298-t006:** ANOVA for the drilling speed (*v*).

Source	DF	Seq SS	Adj SS	Adj MS	*F*-Value	*p*-Value
*U*	1	18.926	18.926	18.926	15.03	0.001
*t_i_*	1	1.357	1.358	1.358	1.08	0.315
*I*	1	36.155	36.281	36.281	28.82	0.000
Residual Error	15	18.884	18.884	1.259	−	−
Total	19	79.874	−	−	−	−

**Table 7 materials-12-02298-t007:** ANOVA for the aspect ratio (*AR*).

Source	DF	Seq SS	Adj SS	Adj MS	*F*-Value	*p*-Value
*U*	1	105.42	105.42	105.422	16.07	0.001
*t_i_*	1	16.01	16.02	16.020	2.44	0.139
*I*	1	145.23	145.63	145.627	22.20	0.000
Residual Error	15	98.38	98.38	6.558	−	−
Total	19	376.35	−	−	−	−

**Table 8 materials-12-02298-t008:** ANOVA for the taper angle (*tap_α_*).

Source	DF	Seq SS	Adj SS	Adj MS	*F*-Value	*p*-Value
*U*	1	0.000242	0.000242	0.000242	6.00	0.027
*t_i_*	1	0.000028	0.000028	0.000028	0.69	0.418
*I*	1	0.000514	0.000515	0.000515	12.76	0.003
Residual Error	15	0.000605	0.000605	0.000040	−	−
Total	19	0.001467	−	−	−	−

**Table 9 materials-12-02298-t009:** ANOVA for linear tool wear (*TW*).

Source	DF	Seq SS	Adj SS	Adj MS	*F*-Value	*p*-Value
*U*	1	5.6667	5.6667	5.6667	2.44	0.139
*t_i_*	1	8.1182	8.2032	8.2032	3.53	0.080
*I*	1	0.5039	0.4945	0.4945	0.21	0.651
Residual Error	15	34.8947	34.8947	2.3263	−	−
Total	19	56.7006	−	−	−	−

**Table 10 materials-12-02298-t010:** ANOVA for the surface roughness parameter *Ra*.

Source	DF	Seq SS	Adj SS	Adj MS	*F*-Value	*p*-Value
*U*	1	0.2244	0.2244	0.2244	1.27	0.277
*t_i_*	1	1.1856	1.1890	1.1890	6.75	0.020
*I*	1	1.9461	1.9437	1.9437	11.04	0.005
Residual Error	15	2.6415	2.6415	0.1761	−	−
Total	19	7.0279	−	−	−	−

**Table 11 materials-12-02298-t011:** ANOVA for the surface roughness parameter *Rz*.

Source	DF	Seq SS	Adj SS	Adj MS	*F*-Value	*p*-Value
*U*	1	25.09	25.20	25.199	2.59	0.130
*t_i_*	1	108.12	108.24	108.240	11.14	0.005
*I*	1	16.96	16.93	16.934	1.74	0.208
Residual Error	14	136.09	136.09	9.721	−	−
Total	19	321.95	−	−	−	−

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
