# Peer review of "Impact of Process Parameters on the Quality of Deep Holes Drilled in Inconel 718 Using EDD"

_materials, 2019, doi:10.3390/ma12142298_

Round 1

Reviewer 1 Report

1.       The abstract should include more information about the methodology, results, and conclusion. For now, it is basically all about background and motivation.

2.       There are typos and broken English throughout the manuscript, for example, on line 5 of abstract, ‘materials are challenge to machine by conventional methods’ should be ‘There are lots of challenges to machine materials by conventional methods’ or ‘It is challenging to machine materials by conventional methods’. One example of typo is the first row of Table 1, it should be ‘output parameters’ instead of ‘out parameters’. Part of the manuscript is difficult to read due to these mistakes which need to be improved.

3.       It is unclear how equations (5) to (10) are derived. It would be helpful to add a Table of original measurements from 20 experiments under different conditions and use a paragraph to explain how the ANOVA analysis is applied.

4.       In Figures 8, 12, it is better to change X axis to the scale of the data value for time of the impulse ti and current amplitude I. For now, all data are evenly distributed although their ti or I are not. Also, the error bars should be added if applicable.

5.       For Figures 5 and 6, the profile of hole is represented in 2D. It is necessary to explain how the diameters are measured. Is it the average diameter throughout the circumference or the diameter at a certain angle? How is the symmetry of drilled holes?

Author Response

Thank you very much for your valuable comments. I have revised the paper accordingly.

Detailed responses to the comments are as follows.

1. The abstract should include more information about the methodology, results, and conclusion. For now, it is basically all about background and motivation.

It is a valuable comment. I have added the more information concerning methodology, results and conclusions  to the abstract:

This paper presents the results of an analysis of the EDD of Inconel 718 alloy. An experiment was conducted to evaluate the impact of process parameters (pulse time, current amplitude, and discharge voltage) on the process’s performance (linear tool wear, taper angle, drilling speed, the hole’s aspect ratio, and surface roughness (Ra and Rz)). The results show that EDD provides us with the possibility to drill holes with an aspect ratio greater than 10:1. The results also demonstrate that the holes with aspect ratio greater than 10:1 and small value of taper angle, have significant decreased quality of internal surface, especially at the hole bottom. It can indicate an that an insufficient amount of debris is removed from the bottom of the hole’.

Added fragment is marked in green colour in the abstract.

2. There are typos and broken English throughout the manuscript, for example, on line 5 of abstract, ‘materials are challenge to machine by conventional methods’ should be ‘There are lots of challenges to machine materials by conventional methods’ or ‘It is challenging to machine materials by conventional methods’. One example of typo is the first row of Table 1, it should be ‘output parameters’ instead of ‘out parameters’. Part of the manuscript is difficult to read due to these mistakes which need to be improved.

The statement materials are challenge to machine by conventional methods’ has been changed to ‘It is challenging to machine materials by conventional methods’.

Sorry for the typing mistake. I have corrected ‘out parameters’ to ‘output parameters’.

Using the MDPI English editing service I made recommended changes in the manuscript.

3. It is unclear how equations (5) to (10) are derived. It would be helpful to add a Table of original measurements from 20 experiments under different conditions and use a paragraph to explain how the ANOVA analysis is applied.

It is a valuable comment. The ANOVA techniques were applied to investigate the relationship between process parameters and input parameters. The equations (5) to (10) are result the obtained analysis of the relationship. I have added the information in the manuscript. Added fragment is marked in green in the manuscript.

Thank you for the suggestion. I have added in the manuscript the Table of original measurements from 20 experiments under different conditions.

4. In Figures 8, 12, it is better to change X axis to the scale of the data value for time of the impulse ti and current amplitude I. For now, all data are evenly distributed although their ti or I are not. Also, the error bars should be added if applicable.

It is a valuable comment. The explanation of the comment is that in case the line (with  the response mean for each factor level connected by a line) is horizontal (parallel to the x-axis), then there is no main effect occur. Each level of the factor affects the response in the same way, and the response mean is the same across all factor levels. When the line is not horizontal, then there is a main effect present. Different levels of the factor affect the response differently. The steeper the slope of the line, the greater the magnitude of the main effect.

It is no possibility to add the error bars because a main effects graphs present the response mean for each factor level (input parameters) influencing  on the output parameter, connected by a line.

5. For Figures 5 and 6, the profile of hole is represented in 2D. It is necessary to explain how the diameters are measured. Is it the average diameter throughout the circumference or the diameter at a certain angle? How is the symmetry of drilled holes?

It is a valuable comment. The way of measurement of diameters is significant. Thank you for the comment. In the manuscript I have added the information concerning the measurements of hole diameter and the symmetry of drilled holes, as follows:

‘The average diameters along the hole’s depth were measured on the two parts of the sample (five measurements were made for each diameter). The measurements were performed using a K-401 stereo microsope with a Common Main Objective (CMO) Infinity optical system and a Moticam 2300 digital camera (Richmond, Canada). The measurements of the diameters were carried out using a MoticImages Plus system. The difference between the appropriate diameters of the two sample parts was in the range 20–30 µm, which leads to assume that the drilled holes were symmetrical’.

Added fragment is marked in green colour in the chapter 2.2 of the manuscript.

Dear Reviewer, thank you very much for your time, professionalism and advices.

Reviewer 2 Report

See attached file

Author Response

Thank you very much for your valuable comments. I have revised the paper accordingly.

The topic of the paper has a relevant interest for the technology.

Thank you !

Detailed responses to the comments are as follows.

1. In general, English must be absolutely improved. A thorough grammar check is mandatory.

A review by an English reader is strongly recommended. 

Thank you for the suggestion. Using the MDPI English editing service I made recommended changes in the manuscript.

2. Please change these words: top diameter instead of input diameter; bottom diameter instead of output diameter; figures instead of scheme; process performance instead of technological factors.

I have changed the words: top diameter instead of input diameter; bottom diameter instead of output diameter; figures instead of scheme; process performance instead of technological factors.

3. The introduction section must be rewritten so that the discussion is more systematic and logical. There is a need to discuss the main motivation of the work and the novelty respect the past papers: what are the shortcomings of past works and how does the proposed paper presented by the author address these shortcomings?

It is a valuable comment. Thank you. I have rewritten the introduction. In the introduction I have added the previously works which deal with the influence of process parameters on the process performance.

I have added in introduction the paragraphs in relation to the main motivation of the work and the novelty:

The phenomenon occurring in machining area between electrodes are still weak recognized what prevents an appropriate selection of process parameter values. The need improvement of the EDD process, especially drilling high aspect ratio holes, is present. Due to the further experimental research should be carried out to improve the efficiency of the process included high material removal rate, low tool wear, satisfied dimensional accuracy and quality of drilled holes’.

and

‘The drilling was carried out at the junction of the two parts of the sample. Once the two parts of the sample were separated, an analysis of the dimensional accuracy and the geometrical characteristics of the internal surface of the holes was carried out. This approach helps us to understand the phenomenon that occurs at the bottom of a hole during the EDD process’.

Added fragments are marked in yellow colour in the introduction of the manuscript.

4. In addition, the process performance is influenced by both the process parameters and the electrodes characteristics (workpiece and electrode). Several papers about this topic are in the literature and should be taken into account.

It is a valuable suggestion. Thank you. I agree that the both process parameters and the electrodes characteristics (workpiece and electrode) have significant influence on the material-removal mechanism during the process. I have considered in the manuscript the suggested articles.

5. Give more details of the experimental research: Rotational speed of the clamp, dielectric pressure, pulse off time, gap, etc.

Thank you for the suggestion. I have added more details concerning the constant parameters, as follows:

‘The following constant parameters were assumed: initial interelectrode gap (S0 = 50 μm), inlet dielectric fluid pressure (pin = 8 MPa), rotational speed of the clamp and the electrode (n = 400 rpm), drilling time of each test (td = 45 min), pulse off time (toff = ti), dimensions and material of the tube electrode (single-channel, outer diameter: 1 mm, inner diameter: 0.3 mm, made of copper) (Figure 3a), and deionized water with a low electrical conductivity as a dielectric fluid. Before each experiment was started, the temperature (T = 297.15–318.15 K) and electrical resistivity (κ = 2–3.9 µS/cm) of the deionized water were measured. The dielectric fluid was flushed down the interior hole of the tube in order to remove eroded particles (Figure 3b)’.

Added fragment is marked in yellow colour in the chapter 2.2 of the manuscript.

I have also added the scheme of the electrical discharge drilling (EDD) process (Figure 3 in the manuscript) in order to explain how working fluid is fed to the machining area.

6. Why is MRR not evaluated? Considering that dimension of the holes is changeable, it could be improve the obtained results.

It is a valuable comment. Thank you. In the presented results of experiments I have analyzed the influence of drilling speed which is related to material removal rate. But the analysis of MRR is more significant in case changeable dimension of  holes.

7. In section “results analysis and discussion”, p-values of the analysis should be added. Figures 8 and 12 report a value for ti=100. Why? In table 1, the range of the impulse is 282-999 μs.

Thank you for the suggestion. The p-values analysis were added in the manuscript.

Sorry for the mistake. The corrected range of the pulse time is 100 – 999 μs.

8. The conclusions should be enhanced such that it provides directions for future work and some recommendation on how the results of the current work can be useful for the researchers about this topic.

Thank you for the comments. I have widened the conclusions as follows:

6.       In those cases where the obtained hole had a conical shape (Din >Dout) but a high aspect ratio (AR > 10), accumulation of debris at the bottom of the hole took place that had an impact on the geometric structure of the hole.

7.       Insufficient flushing efficiency constitutes a limitation of EDD for deep hole drilling. To address this limitation, a special technology that supplies or suctions out working fluid could be applied.

8.       Further experiments should include additional process parameters for the properties of the working fluid, e.g., temperature or density.

Added fragments are marked in yellow colour in the conclusions of the manuscript.

Dear Reviewer, thank you very much for your time, professionalism, advices and recommendations.

Reviewer 3 Report

Manuscript ID: materials- 537523 entitled „Impact of Process Parameters on Quality Drilling Deep Holes in Inconel 718 by Using EDD“ deals with the problem of drilling high aspect ratio holes in Inconel 718 alloy by EDD. The paper describes the highly current EDM problems.

The substantial contents and the description of the manuscript are appropriate and satisfactory. On the whole, I find this paper can be considered for publication in Materials after minor revision.

I have a few questions and remarks about the paper.

1. The all parameters in Table 1 are incorrect described. And also the parameters in some formulas. Need to change.

2.  Mistake in figure title “Figure 3. of sample,...”

3. The paper does not describe electrode shape. Must be added to the paper.

4. How the Ra and Rz parameters were measured? Need to add.

5. Parameter F on page 12 is incorrect. Correctly is S.

Author Response

Thank you very much for your valuable comments. I have revised the paper accordingly.

“The substantial contents and the description of the manuscript are appropriate and satisfactory. On the whole”.

Thank you !

Detailed responses to the comments are as follows.

1. The all parameters in Table 1 are incorrect described. And also the parameters in some formulas. Need to change.

It was corrected. 

2. Mistake in figure title “Figure 3. of sample,...”

Sorry for the typing mistake. Now the Figure 1 is instead of Figure 3 and has title:

 ‘A photograph of the sample, hd—maximal drilling depth (a); the experimental setup of the electrode guiding system (b)’.

3. The paper does not describe electrode shape. Must be added to the paper.

It is a valuable comment. I have added in the paragraph with analysis of linear tool wear the analysis of tool shape:

The measured diameter of the tool tip after drilling was approximately 20% and 10% lower than the nominal value for an applied ti = 550 µs and an applied ti = 999 µs (U = 100 V and I = 3.83 A), respectively’.

Added fragment is marked in blue colour in the chapter 3.1 of the manuscript.

4. How the Ra and Rz parameters were measured? Need to add.

The way of measurement of Ra and Rz parameters is significant. Thank you for the comment. In the manuscript I have added the information how the Ra and Rz parameters were measured as follows:

The average values of surface roughness (Ra and Rz) were measured using a Talysurf Intra 50 profilometer (Taylor Hobson, Leicester, UK). In order to perform the surface roughness measurements, a measuring tip with a rounding radius of 2 µm was used. The measurements were made along the direction of the hole’s depth (parallel to the measuring axis of the hole). A measurement speed of 1 mm/s was used. For the measurements in the two-dimensional (2D) system, the resolution of the X axis was equal to 1 µm, and five elementary sections 0.8 mm in length were applied’.

Added fragment is marked in blue colour in the chapter 2.2 of the manuscript.

5. Parameter F on page 12 is incorrect. Correctly is S.

It was corrected.

Dear Reviewer, thank you very much for your time, professionalism, advices and recommendations.

Round 2

Reviewer 1 Report

Comments are addressed properly and the manuscript can be accepted in present form.

Reviewer 2 Report

I can confirm that my main concerns raised in the first review have been answered by the proposed modifications.